# Physiotherapy and Exercise Management of People Undergoing Surgery for Lung Cancer: A Survey of Current Practice across Australia and New Zealand

**DOI:** 10.3390/jcm12062146

**Published:** 2023-03-09

**Authors:** Georgina A. Whish-Wilson, Lara Edbrooke, Vinicius Cavalheri, Linda Denehy, Daniel Seller, Catherine L. Granger, Selina M. Parry

**Affiliations:** 1Department of Physiotherapy, School of Health Sciences, The University of Melbourne, Melbourne, VIC 3010, Australia; 2Department of Health Services Research, Peter MacCallum Cancer Centre, Melbourne, VIC 3000, Australia; 3Curtin School of Allied Health, Curtin University, Perth, WA 6845, Australia; 4enAble Institute, Faculty of Health Sciences, Curtin University, Perth, WA 6845, Australia; 5Allied Health, South Metropolitan Health Service, Perth, WA 6009, Australia; 6Physiotherapy, ORA Therapies, Capital & Coast District Health Board, Wellington 6242, New Zealand; 7Department of Physiotherapy, The Royal Melbourne Hospital, Melbourne, VIC 3050, Australia

**Keywords:** lung cancer, physiotherapy, exercise, thoracic surgery, cross-sectional survey

## Abstract

Background: Moderate- to high-certainty evidence supports the benefits of pre- and post-operative exercise for people undergoing surgical resection for lung cancer. Despite this, exercise programs are not commonly provided. Previous data regarding exercise practices are a decade old. Therefore, this study aimed to understand current exercise practices in surgical lung cancer care in Australia and New Zealand. Methods: An online cross-sectional survey of Australian and New Zealand allied health professionals specialising in exercise-based interventions was carried out. Survey development and reporting adhered to CHERRIES and CROSS checklists. Institutions with thoracic surgery departments were invited to participate via email, and additional responses were sought via snowballing. Results: The response rate was 81%, with a total of 70 health services responding. A total of 18 (26%) pre-operative services, 59 (84%) inpatient post-operative services, and 39 (55%) community/outpatient post-operative services were identified. Only eight (11%) services provided a pre-operative exercise program. Half of the respondents referred less than 25% of patients to community/outpatient exercise programs on hospital discharge. Respondents reported that their clinical management was predominantly influenced by established workplace practices and personal experience rather than evidence. Conclusions: The availability and uptake of pre- and post-operative exercise remain low, and work should continue to make pre/post-operative exercise training usual practice.

## 1. Introduction

The optimal curative treatment for early-stage lung cancer is surgical lung resection [1]. In the first 12 months post-operatively, many patients experience increased pain, shortness of breath and fatigue, and decreased peripheral muscle strength, functional capacity, and health-related quality of life (HRQoL) [2,3,4]. Physiotherapists and other exercise health professionals are well placed to play a vital role in the management of these post-operative sequelae, primarily through the provision of pre- and post-operative exercise programs, with moderate- to high-certainty evidence supporting the benefits of such programs in this population [5,6]. 

Despite this, it is uncommon in Australia, New Zealand, and elsewhere in the world for people undergoing lung cancer surgery to be referred to pre- or post-operative exercise programs [3]. Typically, exercise services are currently only routinely offered to people undergoing surgical resection for lung cancer in the form of early inpatient post-operative physiotherapy, generally focused on respiratory management and facilitating discharge home [3]. The most recent data on current practice in this area were published a decade ago, reporting low uptake of both pre- and post-operative exercise programs for people with lung cancer in Australia and New Zealand [7]. Aside from exercise, other common physiotherapy techniques utilised in lung cancer management include but are not limited to cardiorespiratory techniques (often referred to as ‘chest physiotherapy’, e.g., airway clearance and breathing techniques) and education.

It is pertinent now to update our understanding of current exercise health professional practice in this area, given the newly available breadth of evidence, improved patient survival rates [8], and changes to surgical management (e.g., increased uptake of minimally invasive procedures [9]) over the last decade. Anecdotally, exercise health professionals working in this area report that clinical practice has not changed in line with the growing evidence base. By first gaining an accurate picture of current clinical practice, it will then be possible to identify potentially modifiable barriers hindering implementation, highlighting key areas to target in the pursuit to integrate exercise into routine clinical practice. 

The aims of our study were to (1) identify the current role of exercise health professionals/clinicians in the management of people undergoing surgical resection for lung cancer across the continuum of care (pre-operative, post-operative, and community/outpatient) in Australia and New Zealand and (2) investigate the success in implementing the evidence base into routine clinical practice over the last decade. 

## 2. Materials and Methods

### 2.1. Study Design

A purpose-built, cross-sectional, open online survey was developed using the online survey platform Qualtrics [10] and pre-tested for usability. 

The survey was developed by adapting the questionnaire by Cavalheri and colleagues from 2013, which examined management from the perspective of physiotherapists only across Australia and New Zealand [7]. The survey was expanded to include the perspectives of other exercise clinicians (e.g., exercise physiologists and allied health assistants) and those working in timepoints along the continuum beyond the inpatient post-operative period (e.g., pre- and post-operative in- and outpatient programs). The survey was divided into three main sections: (1) eligibility screening (6 questions), (2) demographics (14 questions), and (3) current clinical practice questions focused on elements of assessment, treatment, and education (32–34 questions per timepoint (pre-operative, post-operative, and community/outpatient post discharge)). The eligibility section of the survey included the participant information and consent form and required informed consent (tick box response) prior to commencement of the survey. This provided respondents with information relating to the time commitment of the survey, which data were stored and where/for how long, who the investigators were, and the purpose of the study. 

Respondents could self-select their primary area(s) of clinical practice and were then directed to complete the relevant section(s) using branching logic. This enabled adaptive questioning whereby only certain items or sections were displayed based on responses to other items, reducing the number and complexity of questions. Closed question formats included drop-down responses, binary yes–no, multiple choice, and 5-point Likert scales. Some questions allowed the selection of multiple responses (i.e., all that apply). Additionally, participants were asked to rate the factors that influenced their assessment and treatment of people with lung cancer undergoing surgical resection. These factors were based on the initial 2013 survey [7]. All items except binary responses provided respondents an option to provide additional “other” responses. An example of the survey questions for a respondent working in the pre-operative setting can be found in Appendix A.

A three-stage testing process was implemented prior to dissemination to confirm the face validity, usability, and technical function of the electronic survey; one author (GWW) designed the online survey, following which the remaining authors reviewed and refined the survey until consensus was reached. The survey was then tested twice with a representative pilot sample of clinician researchers (n = 3 physiotherapists working in private/public hospitals, all different levels of seniority and specialisation) who provided feedback on the survey clarity, ease of completion, and content to confirm the face validity and usability of the survey. This step was crucial to enhance validity, improve responder reliability, and help reduce potential measurement error or non-response errors in the live survey. Minor revisions to wording for clarification and technical elements (e.g., branching logic) were made to the survey based on feedback from pilot testing. 

Data were stored in a secure, password-protected, University-hosted server accessible only to the research team. Data were de-identified prior to analysis, and all re-identifiable data were stored separately from the main data file. 

The survey was open for five weeks between June and July 2022. Incomplete surveys were automatically submitted after survey closure. As an incentive to participate in the study, respondents were eligible to enter a draw to win a new Apple iPad upon completing the survey. This was facilitated through a separate process that could not be linked to their survey answers. 

The development and reporting of the survey were informed by the Checklist for Reporting Results of Internet E-Surveys (CHERRIES) and Checklist for Reporting of Survey Studies (CROSS) (Appendix A) [11,12]. 

### 2.2. Participants and Recruitment

Eligibility was assessed in the opening pages of the survey. Clinicians were eligible to participate if they were: Allied health professionals specialising in exercise-based interventions (e.g., physiotherapists, exercise physiologists, and allied health assistants);Currently working in a setting that manages people undergoing surgical resection for lung cancer, in any setting across the continuum (e.g., pre-operative, inpatient post-operative, and community/outpatient post discharge) to ensure data collected were reflective of current practices;Based in Australia or New Zealand.

Respondents were excluded if they did not progress beyond the demographics section (i.e., if they provided no data on clinical practice). No further exclusion criteria were defined. Respondents were asked to nominate their name, workplace, and professional email address to ensure response accountability and prevent multiple entries from the same individual. 

Hospitals with thoracic surgery services were identified as the main target for dissemination. A list of hospitals in Australia and New Zealand that likely provided thoracic surgery services was created through interstate and international clinical networks of investigators and through Internet searches and was cross-checked against the list used in a previous survey [7]. The physiotherapy and/or allied health managers of identified health services were emailed the survey link directly and asked to disseminate the survey to eligible clinicians. It was asked that only one respondent from each eligible discipline and timepoint completed the survey. Invitations were emailed to all identified services at the same time, following which reminder emails were sent on two occasions (two weeks and one week prior to survey closure) to any health services that had not commenced the survey and to respondents who had commenced but not completed the survey.

Additional responses were sought via online snowball sampling. The survey was advertised via multiple avenues including collaborative networks and social media (Figure 1). An example of the social media survey announcement is published in Appendix A. 

### 2.3. Management of Responses 

Where duplicate responses were received (e.g., multiple responses from the same health service, discipline, and site), a hierarchy was used to determine which response would be included in the analysis of clinical practice data. Firstly, the most complete response was prioritised. If all duplicate responses were complete, the response from the most senior clinician as determined by years of clinical experience was included in the analysis. Duplicate responses were included in the analysis of demographic data to avoid over-estimating the seniority of clinicians working in this area. 

Incomplete responses were included in the data analysis. Ineligible responses (responses that did not meet eligibility criteria) were removed prior to analysis.

### 2.4. Data Analysis 

De-identified survey data were exported, coded, and analysed descriptively using Statistical Package for Social Sciences (SPSS) Version 28. Demographic and clinical practice data are displayed as either n (%) for categorical variables, median [IQR], or mean (SD) depending on the distribution of the data using the Kolmogorov–Smirnov test of normality. Responder rurality was coded according to the Australian Statistical Geography Standard (ASGS) [13] and the New Zealand Geographic Classification for Health [14]. Demographic differences between responders and non-responders were compared using Fisher’s exact test.

The response rate was defined as the number of institutions returning a response, divided by the number of institutions invited to participate via email. The completion rate was defined as the number of submitted surveys (irrespective of completeness), divided by the number of users who consented to participate (i.e., the percentage of respondents who progressed beyond the informed consent stage) [11]. The completeness rate was defined as the number of surveys that were 100% complete, divided by the number of surveys submitted [11].

## 3. Results

Between June and July 2022, 97 potentially eligible health services were identified, and contact was attempted (Figure 1). The survey was subsequently administered to the 70 health services confirmed as eligible, 57 of which provided an eligible response for at least one timepoint (response rate of 81% n = 57/70). Three ineligible responses were excluded prior to analysis due to servicing the wrong population (n = 2) and not providing a valid workplace email address (n = 1).

A comparison of responder (n = 57) and non-responder (n = 13) demographics revealed a higher proportion of private health services within the non-responder group (n = 9/13 (69%) versus responders n = 22/57 (39%), *p* = 0.030). No difference was observed in the proportion of rural/remote health services between the groups (non-responders n = 3/13 (23%) versus responders n = 3/57 (5%), *p* = 0.073). 

Thirteen additional health services were identified via snowballing, all of which were eligible to participate and responded. Overall, from across the 70 responding health services, a total of 132 participants consented to participate in the survey, and 102 valid responses were received. The flow of survey responses from health services is summarised in Figure 1.

The survey completion rate was 77% (n = 102/132) (i.e., 30 respondents provided informed consent but did not progress to participate in the survey). Of those who progressed past the informed consent stage, the survey completeness rate was 85% (n = 87/102) (i.e., 15 respondents did not complete the entire survey). No responses were excluded based on completeness. Nine duplicate responses were excluded from the analysis of clinical data. 

### 3.1. Demographics of Respondents 

One hundred and two clinicians responded to the survey (Table 1). Most (n = 90; 88%) were physiotherapists, 11 (11%) were exercise physiologists, and 1 (1%) was an allied health assistant. Their most common primary work setting for clinical practice with people undergoing surgical resection for lung cancer was the inpatient post-operative setting (n = 65, 64%). Thirty-two (46%) institutions offered exercise services at more than one timepoint across the continuum. 

### 3.2. Demographics of Identified Health Services 

Seventy unique health services (Table 1) responded from across all states/territories in Australia and both islands of New Zealand. Only seven (10%) were rural and/or remote health services. The spread of respondent discipline and health service rurality across the continuum is demonstrated in Figure 2. 

### 3.3. Lung Cancer Management 

The current clinical practice (assessment and treatment) of people undergoing surgical resection for lung cancer is summarised in Figure 3, Figure 4, Figure 5 and Figure 6 across pre-operative, post-operative, and community/outpatient settings, respectively. The outcome measures used by clinicians are summarised in Appendix A. Key differences and similarities in exercise providers, education, and prescription across the different timepoints are summarised in Appendix A. Established workplace practices and personal experience had the most influence on clinicians’ management of people with lung cancer (Table 2).

### 3.4. Pre-Operative

Eighteen (26%) health services had a pre-operative exercise service for people awaiting surgical resection for lung cancer. Two (11%) of the pre-operative services were in rural/remote areas. The most common discipline who responded that they were working in the pre-operative setting was physiotherapists (n = 14, 78%), and the remaining four (22%) respondents working in this setting were exercise physiologists. Of the 18 identified services, 6 (33%) were exercise/prehabilitation programs, 6 (33%) were exercise services based within pre-admission clinics, 5 were inpatient ward-based exercise services (28%), and 1 (6%) was exclusively a pre-operative exercise capacity assessment.

Six (33%) and five (29%) services reported that ‘most’ or ‘some’ patients were assessed by exercise clinicians prior to surgery, respectively. Twelve services (67%) reported patient-related factors associated with triggering pre-operative assessment. The most common triggers were respiratory comorbidity (n = 9, 50%), frailty (n = 9, 50%), advanced age (n = 6, 33%), and poor performance in functional tests (n = 6, 33%). Other factors influencing whether a pre-operative assessment occurred included referral processes (n = 12, 66%), staff availability (n = 11, 61%), and patient availability (n = 5, 28%). The parameters assessed by clinicians pre-operatively are summarised in Figure 3a, and the most common outcome measures used to assess each parameter are summarised in Appendix A.

All services provided pre-operative education delivered by an exercise health professional, with nine (50%) services providing education to ‘all’ or ‘most’ patients. Education topics are summarised in Figure 3b.

Eight services (44%) reported that at least ‘a few’ patients participate in pre-operative exercise, inclusive of five prehabilitation services, one pre-admission clinic service, and two inpatient ward-based services. No rural/remote pre-operative services offered exercise programs. Pre-operative exercise programs ran for a median of 3 [1.5–4] weeks and provided three [2–5] sessions per week, and individual sessions ran for a median of 60 [30–60] min. Characteristics and components of exercise programs are summarised in Figure 3c,d.

The most common non-exercise components of pre-operative services included nutritional counselling (n = 8, 44%), smoking cessation (n = 7, 39%), and psychological support (n = 7, 39%).

### 3.5. Inpatient Post-Operative

Fifty-nine (84%) health services had an inpatient post-operative exercise service for people undergoing surgical resection for lung cancer. Four (7%) of the inpatient post-operative services were in rural/remote areas. All primary respondents were physiotherapists, with one duplicate response from an allied health assistant. Thirty-one (53%) respondents reported that, of the patients they are involved with post-operatively, no patients participated in pre-operative exercise, with 15 (25%) and three (5%) reporting that ‘a few’ patients or ‘some’ patients participated in pre-operative exercise, respectively.

Most respondents (n = 37, 63%) reported the presence of a standard clinical pathway in the post-operative management of these patients, and most included early post-operative physiotherapy assessment and early mobilisation. Forty-three (73%) services reported a blanket referral system for physiotherapy for all patients with lung cancer undergoing thoracic surgery. In health services without blanket referrals, the most common triggers for physiotherapy referral were open surgery (e.g., thoracotomy) (n = 7, 43%), risk of post-operative pulmonary complication (n = 4, 25%), and concerns regarding mobility/safety on discharge (n = 4, 25%).

Post-operative physiotherapy input usually commenced on the first post-operative day (n = 55, 93%), with two (3%) reporting commencing input on the day of surgery. Across health services, ‘most’ (n = 16, 27%) or ‘all’ patients (n = 40, 68%) were assessed post-operatively by physiotherapists. The most common parameters assessed by physiotherapists were cardiorespiratory status (n = 59, 100%), mobility/physical function (n = 58, 98%), and pain (n = 57, 97%). Assessments utilised by physiotherapists at this timepoint are summarised in Figure 4a, and the most common outcome measures used to assess each parameter are summarised in Appendix A.

Most respondents reported that ‘all’ (n = 39, 67%) or ‘most’ (n = 15, 26%) patients received some form of post-operative intervention from clinicians, including post-operative education and/or exercise-based interventions. Education topics and exercise interventions commonly prescribed are summarised in Figure 4b,c. Post-operative physiotherapy sessions ran for a median of 30 [21–30] min and were provided a median of one [1–1.5] time per day.

Referral rates to post-operative community/outpatient exercise programs on discharge from hospital were low (Figure 5). Most respondents reported referring either ‘less than 25%’ (n = 23, 40%) or ‘no’ patients (n = 6, 10%), with only 11 services (19%) referring more than 50% of patients. Community/outpatient referral rates were lower in rural/remote areas, with 66.7% (n = 2) reporting referring ‘less than 25%’ of patients. Common triggers for referral were reduced exercise tolerance (n = 41, 79%), respiratory comorbidity (e.g., chronic obstructive pulmonary disease, COPD) (n = 31, 60%), and new gait aid requirement (n = 20, 38%). The services they were referred to were mainly pulmonary rehabilitation (n = 40, 77%) or oncology rehabilitation programs (n = 13, 25%), and referrals were usually completed by physiotherapists (n = 47, 90%).

### 3.6. Community/Outpatient

Thirty-four (49%) health services had a community/outpatient post-operative exercise service for people after surgical resection for lung cancer. Five health services (7%) (all metropolitan) reported having multiple community/outpatient post-operative services (total services identified n = 39). Four (12%) community/outpatient post-operative services were in rural/remote areas. A total of 77% (n = 30) of respondents for this timepoint were physiotherapists, and the remaining 23% (n = 9) were exercise physiologists. This timepoint, therefore, represents the most common timepoint of exercise physiology involvement. The most common community/outpatient exercise service type was pulmonary rehabilitation (n = 27, 69%) followed by oncology rehabilitation (n = 15, 38%).

The time between hospital discharge and commencing in community/outpatient exercise services varied. Most commonly, patients commenced in programs after longer than 8 weeks (n = 14, 36%). Six (15%) services reported patients commencing within 4 weeks of hospital discharge. One hundred percent (n = 4) of rural/remote services reported wait times longer than 8 weeks. Wait times did not differ based on the type of program.

The parameters assessed by clinicians at the community/outpatient post-operative timepoint are summarised in Figure 6a. The most common assessment tools utilised are summarised in Appendix A.

Seventeen (45%) and eleven (29%) services reported that ‘all’ or ‘most’ patients receive post-operative education from an exercise health professional, respectively. Education topics are summarised in Figure 6b.

Most respondents reported that ‘all’ (n = 14, 37%) or ‘most’ (n = 15, 40%) patients participate in an exercise program during this timepoint. Post-operative exercise programs ran for a median of eight [8-8] weeks, provided two [2-2] sessions per week, and ran for 60 [45–60] min. The characteristics and components of exercise programs are summarised in Figure 6c,d. Twenty-one (62%) respondents reported that ‘most’ patients complete the exercise program as prescribed.

Referral rates to longer-term maintenance exercise programs on completion of post-operative community/outpatient programs were variable. Eight (24%) and five (15%) respondents reported that either ‘less than 25%’ or ‘no’ patients were referred onwards on completion, respectively, whereas six (18%) and two (6%) respondents reported referring either ‘more than 75%’ or ‘100%’ of patients, respectively.

## 4. Discussion

This study provides in-depth insight into the current exercise management of people undergoing surgical resection for lung cancer. In line with previous studies, our results suggest that most exercise professional input occurs in the initial inpatient post-operative phase. Other key findings of this survey include the following: (1) only 18 (26%) health services reported having a pre-operative service, 8 (44%) of which included an exercise program; (2) half of the respondents working in the inpatient post-operative setting reported referring less than 25% of patients to exercise programs on hospital discharge; and (3) only 34 (49%) health services reported having a community/outpatient post-operative exercise service.

Despite moderate- to high-certainty evidence supporting the efficacy of pre- and post-operative exercise for patients with lung cancer, such programs are currently scarcely available across Australia and New Zealand (particularly in rural and remote areas), and referral to programs is still not routine practice. Our findings suggest a potential small improvement in the availability of, and referral to, post-operative exercise services in comparison to the 2013 study, which reported that 72% (compared to 50% in the present study) of respondents referred less than 25% of patients to post-operative exercise services [7]. The availability of pre-operative services appears to be largely unchanged (91% of hospitals did not incorporate pre-operative exercise training [7], compared to 86% in the present study). Several international guidelines support the routine provision and embedding of pre- and post-operative exercise services along the continuum of cancer care irrespective of cancer type [15,16,17], including Enhanced Recovery After Surgery (ERAS) guidelines [18,19], and moderate- to high-certainty evidence supports this among the surgical lung cancer population [5,6]. Our findings suggest that in Australia and New Zealand, these guidelines have not yet successfully been integrated into clinical practice. This persistent gap between evidence and clinical practice is perhaps partially explained by our finding that clinicians reported established workplace practice and personal experience had more influence on their management of people undergoing surgical resection for lung cancer than journal articles.

Our study identified a number of potential barriers to the implementation of exercise into routine lung cancer care, some of which have been identified previously in the literature, including clinician knowledge, workplace culture, constraints of the health care system (e.g., clinician time, staffing, and protocols), and patient appointment burden [20,21,22].

Regarding the implementation of pre-operative exercise, several potentially modifiable barriers were identified. Anecdotally, professional input about exercise during the pre-admission clinic visit is often ad hoc and reliant on referrals from medical staff. Traditionally, patients are scheduled to see multiple clinicians on the day of the pre-admission clinic visit (e.g., surgeons, nursing staff, anaesthesiology, radiology, and pathology), and rarely are exercise health assessments pre-planned. The ad hoc nature of these appointments presents logistical barriers such as referral processes and staff/patient availability. By pre-scheduling exercise professional assessments during lung cancer pre-admission clinic visits, as is routinely carried out for other disciplines and has been shown to be beneficial in other surgical populations [23], these challenges could potentially be circumvented. This may also reduce patient appointment burden and could lead to an increased number of patients referred to pre-operative exercise programs. The speed at which patients typically proceed from lung cancer diagnosis to surgery (often within two weeks [9]) is another often cited barrier to implementation. A recent Cochrane review, however, found no difference in benefits sustained from pre-operative exercise programs that ran for two weeks or less compared to three–four-week-long programs [6]. Should patients be identified and referred early after diagnosis, perhaps through the integration of an automatic referral system, it is possible that most patients could successfully complete an effective pre-operative exercise program. Input at this timepoint may be currently limited by funding and personnel availability within physiotherapy/allied health departments, and further studies investigating the cost-effectiveness of pre-operative interventions may strengthen the business case for increased input at this timepoint. Increased access to pre-operative exercise programs may also improve access to curative surgical treatment for lung cancer for patients deemed unfit for surgery due to physical fitness and/or functional performance limitations and/or assist in preparing/optimising people with lung cancer scheduled to undergo subsequent surgeries [3,24].

Aside from early mobility (which is typically prescribed to optimise post-operative respiratory status), clinicians working in the post-operative inpatient timepoint reported relatively low rates of both exercise prescription and education regarding exercise guidelines. Given that in 63% of health services, post-operative exercise management followed a standard post-operative clinical pathway, it is likely that existing pathways/protocols do not routinely include exercise prescription or education. Given these findings, and the strong influence of personal experience and established workplace practice on clinical decision making, it may be warranted to conduct further research into both (1) the contents and ongoing validity of current post-operative clinical pathways and (2) individual clinicians’ knowledge, attitudes, and beliefs regarding exercise prescription for patients undergoing lung cancer surgery to identify any further barriers to implementation.

Additionally, while no research exists suggesting that subgroups of people with lung cancer benefit more from exercise programs, our findings suggest that patient factors such as respiratory comorbidity (e.g., COPD) and impairment (e.g., shortness of breath and mobility restriction) are commonly used by clinicians as indicators for referrals to post-operative exercise. These findings are likely explained by the longstanding routine practice of referring patients with COPD to pulmonary rehabilitation programs and those with post-operative impairments to community rehabilitation programs. Given the relatively new breadth of evidence supporting exercise for all people undergoing surgical resection for lung cancer, interventions targeting clinician awareness of the evidence may be warranted.

Clearly, the implementation of exercise into routine clinical practice would be helped by increasing the number of services available and by ensuring the availability of remote (e.g., telehealth) delivery alongside face-to-face models, particularly in rural and remote areas where service availability is significantly lower and lung cancer incidence is higher [8]. Aside from increasing the actual number of services, which is likely infeasible in the short term given the constraints of the health care system, it is important that current exercise oncology services expand their eligibility criteria to include people with lung cancer. A previous survey of oncology rehabilitation providers suggested that only 12% of pre-treatment programs and 54% of post-treatment programs accepted referrals for people with lung cancer (data include both surgical and non-surgical populations) [25]. No such data are available regarding pulmonary rehabilitation. Accepting referrals for this population would undoubtedly improve access and equity.

Some strengths of our study include the high response rate (81%) with representation across all states, territories, and regions in Australia and New Zealand. This means we have strong confidence that the results are reflective of current practices amongst exercise health services across Australia and New Zealand. However, as in the 2013 survey (which reported that 100% of non-responders were private health services [7]), we observed an increased proportion of private health services within the non-responder group, which may have influenced our results. No difference was observed between the rurality of responders and non-responders. Recall bias was minimised by only including clinicians currently working in lung cancer care and encouraging health services to distribute the survey to the clinicians(s) most involved with this population. Our study also had limitations. Due to the nature of our sampling approach, it is possible that sites providing exercise care to patients with lung cancer may have been missed. For example, we did not purposefully seek to advertise the survey to private practice/community settings. However, the majority of lung cancer surgical care typically occurs via the acute hospital system. Additionally, the COVID-19 pandemic may have influenced responses, and therefore, the current practice may not be reflective of typical pre-pandemic routine practice. For example, at the time of the survey, some facilities had not yet resumed thoracic surgical services, and some community/outpatient services were suspended during the pandemic. As respondents were required to currently be working with patients with lung cancer to be eligible, sites with suspended services were ineligible to complete the survey. Additionally, it was not possible within the boundaries of an online survey to acknowledge the often complex and multifaceted factors that may influence the rates of exercise prescription and referral in this population. Further qualitative research, for example, with focus groups or semi-structured interviews, could investigate these issues in greater depth. Finally, as this study only included the perspectives of Australian and New Zealand clinicians, the international applicability of our findings is unclear, particularly in countries with privately funded health care systems.

## 5. Conclusions

The results of this study show a clear, persistent gap between research evidence and exercise professional clinical practice in surgical lung cancer care. Despite moderate- to high-certainty evidence supporting the routine use of pre- and post-operative exercise, such services remain scarce, and referral rates remain low. Work should continue to integrate exercise into the pathway of lung cancer surgical care, including further research to understand clinician knowledge and beliefs relating to exercise in this population and the implementation enablers and barriers they may face when attempting to translate research into practice.

## Figures and Tables

**Figure 1 jcm-12-02146-f001:**
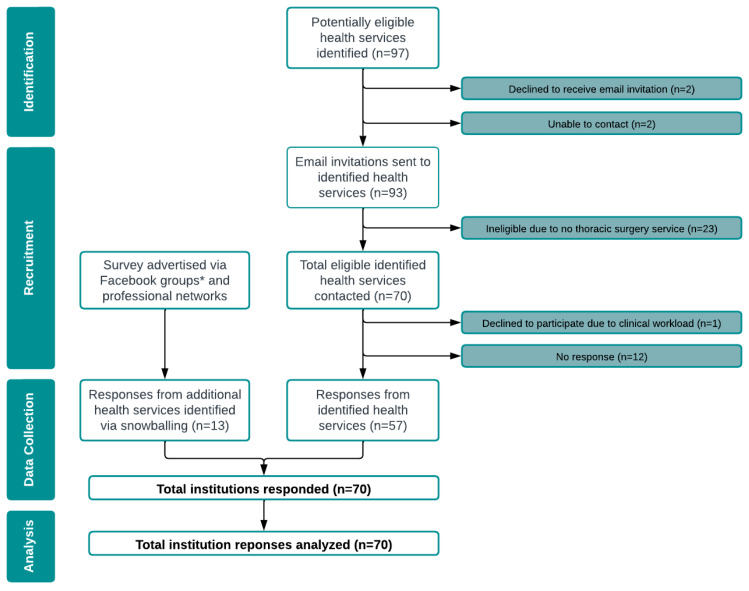
Flow of invited and snowballed health services throughout the study. * Australian Physiotherapy Association (APA) Cardiorespiratory Group; APA Cancer, Palliative Care and Lymphoedema Group; Exercise and Sport Science Australia (ESSA) Cardiovascular Group; ESSA Cancer Group; and Pulmonary Rehabilitation Network.

**Figure 2 jcm-12-02146-f002:**
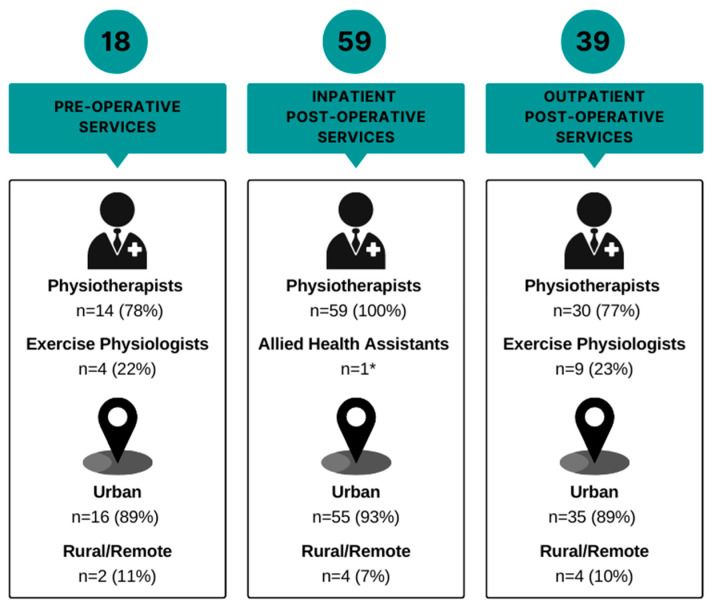
Distribution (number and percentage) of respondent discipline and health service rurality across the continuum of care. * Allied health assistant response received in addition to physiotherapist response from a health service.

**Figure 3 jcm-12-02146-f003:**
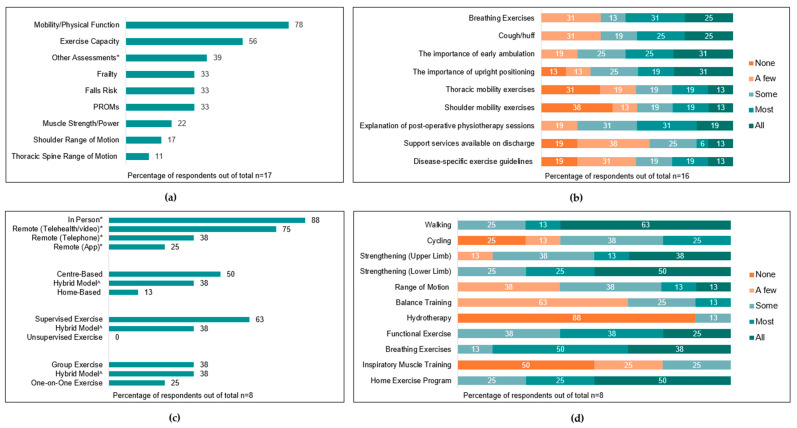
Pre-operative assessment and management of people awaiting surgical resection for lung cancer (%). (**a**) Types of assessments performed (PROMs = patient-reported outcome measures; * other answers provided: lung function testing, n = 7); (**b**) proportion of patients with lung cancer receiving education on certain topics; (**c**) modes of exercise program delivery (* respondents could select >1 option; ^ hybrid refers to a combination of both options (e.g., both centre- and home-based)); (**d**) proportion of patients participating in exercise types (other answers provided: stretching, n = 1).

**Figure 4 jcm-12-02146-f004:**
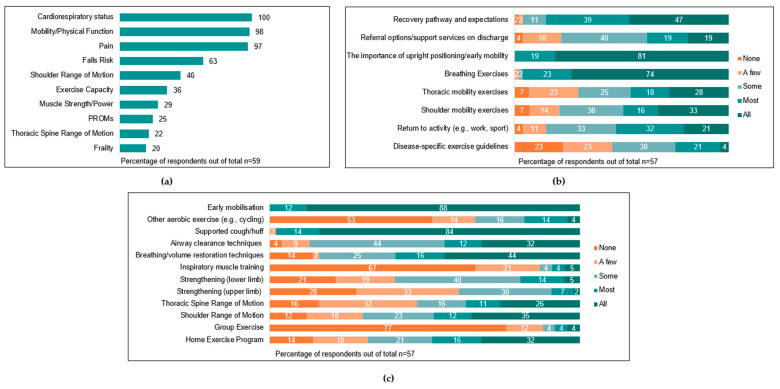
Inpatient post-operative assessment and management of people post surgical resection for lung cancer (%). (**a**) Types of assessments performed (PROMs = patient-reported outcome measures); (**b**) proportion of patients with lung cancer receiving post-operative education on certain topics (‘other’ answers provided: lifting advice, n = 1; pain management, n = 2); (**c**) proportion of patients participating in different exercise types post-operatively (‘other’ answers provided: core exercise, n = 1; balance exercises, n = 1).

**Figure 5 jcm-12-02146-f005:**
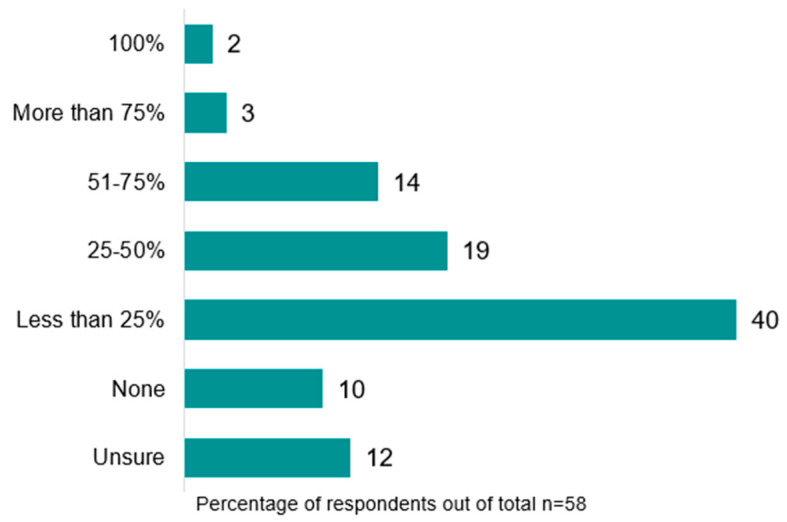
The proportion of people undergoing surgical resection for lung cancer referred to community/outpatient post-operative exercise programs on hospital discharge (%).

**Figure 6 jcm-12-02146-f006:**
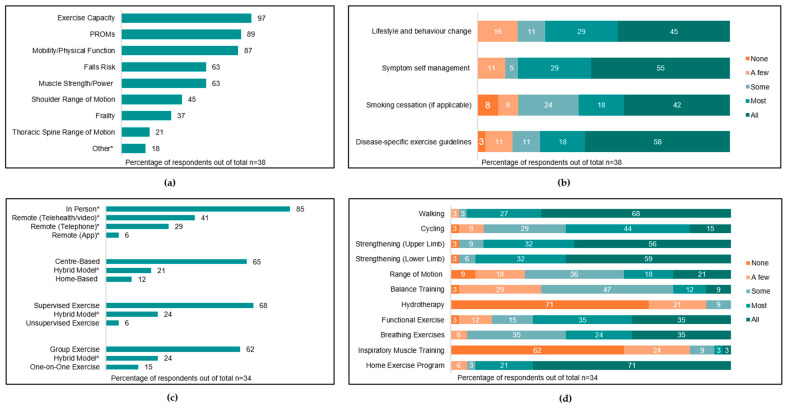
Community/outpatient assessment and management of people post surgical resection for lung cancer (%). (**a**) Types of assessments performed (PROMs = patient-reported outcome measures; *other answers provided: lung function testing, n = 7); (**b**) proportion of patients with lung cancer receiving education on certain topics (other answers provided: nutrition, n = 3; breathlessness, n = 2; airway clearance, n = 1; exercise progression, n = 1; journal articles, n = 1; psychosocial, n = 1; and scar management, n = 1); (**c**) modes of exercise program delivery (* respondents could select >1 option; ^ hybrid refers to a combination of both options (e.g., both centre- and home-based)); (**d**) proportion of patients participating exercise types (* other = ACBT, n = 1; pain management, n = 1; pilates, n = 1; pilates/HIIT/circuit/boxing/arm ergo, n = 1; stretching/relaxation/Tai Chi, n = 1; and treadmill, n = 1).

**Table 1 jcm-12-02146-t001:** Characteristics of responding clinicians and health services.

Demographics of Clinician Respondents (n = 102)	n (%) or Median [IQR]
Sex	
Female	71 (70)
Male	31 (30)
Discipline	
Physiotherapist	90 (88)
Exercise Physiologist	11 (11)
Allied Health Assistant	1 (1)
Other	0
Highest level of education completed	
Workforce entry diploma/certificate	2 (2)
Workforce entry degree	76 (75)
Post-graduate coursework specialisation	10 (10)
Research master’s degree	8 (8)
Research doctorate degree	4 (4)
Other ^a^	2 (2)
Country of workforce entry qualification	
Australia	88 (86)
New Zealand	3 (3)
United Kingdom	6 (6)
Other ^b^	5 (5)
Years working in discipline	
1–5 years	18 (18)
6–10 years	31 (30)
11–15 years	21 (21)
More than 15 years	32 (31)
Years working in area of lung cancer	
Less than 1 year	4 (4)
1–5 years	41 (40)
6–10 years	34 (33)
11–15 years	3 (3)
More than 15 years	20 (20)
Primary timepoint of contact with lung cancer patients ^c^	
Pre-Operative	6 (6)
Acute inpatient post-operative	65 (64)
Community/outpatient post-operative	31 (30)
Frequency of referrals received for lung cancer patients	
Very often (once a week or more)	41 (40)
Often (once a fortnight)	27 (27)
Sometimes (once a month)	25 (25)
Rarely (once every 6 months)	8 (8)
Very rarely (once a year or less)	1 (1)
% work week allocated to clinical activities	80 [67.5–85]
**Characteristics of health services (n = 70)**	
Location	
Victoria (AUS)	19 (27)
New South Wales (AUS)	16 (23)
Queensland (AUS)	14 (20)
South Australia (AUS)	3 (4)
Western Australia (AUS)	3 (4)
Tasmania (AUS)	2 (3)
Australian Capital Territory (AUS)	1 (1)
Northern Territory (AUS)	1 (1)
North Island (NZ)	6 (9)
South Island (NZ)	5 (7)
Service Rurality	
Urban	63 (90)
Rural and remote	7 (10)
Service Funding	
Public	44 (63)
Private	24 (34)
Other ^d^	2 (3)
Lung cancer exercise services reported	
Pre-Operative	18 (26)
Acute inpatient post-operative	59 (84)
Community/outpatient post-operative	39 (52) ^e^

^a^ Bachelor of Science (n = 2); ^b^ South Africa (n = 2), United Arab Emirates (n = 1), Ireland (n = 1), and The Netherlands (n = 1); ^c^ one selection per participant; ^d^ university health clinic (n = 1) and public–private partnership hospital (n = 1); and ^e^ n = 75 due to the inclusion of 5 additional services identified within duplicate organisations (e.g., two unique community/outpatient services offered by the same organisation).

**Table 2 jcm-12-02146-t002:** Factors influencing respondent management of people with lung cancer.

Factor	Response Options, n (%)				
	Not at All	A Little	Somewhat	A Lot	Very Much
Published journal articles	1 (1)	14 (14)	38 (37)	38 (37)	11 (11)
Textbooks	16 (16)	41 (40)	35 (34)	9 (8.8)	1 (1)
Established workplace practices	2 (2)	3 (2.9)	19 (19)	48 (47)	30 (29)
Personal experience	0	5 (4.9)	18 (18)	55 (54)	24 (24)
Postgraduate education	16 (16)	17 (17)	28 (28)	31 (30)	10 (9.8)
Professional development	4 (3.9)	13 (13)	32 (31)	39 (38)	14 (13)
Workforce entry degree	15 (14)	34 (33)	34 (33)	15 (15)	4 (3.9)

Other influences provided (n = 20): surgeon preference (n = 2) working with other clinicians (n = 8), patient preference (n = 2), practice guidelines (n = 2), and clinical reasoning, empathy, working in a tertiary training hospital, and participating in research activities (all n = 1).

## Data Availability

The data are not publicly available due to privacy and ethical restrictions.

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
