# Peer review of "Physiotherapy and Exercise Management of People Undergoing Surgery for Lung Cancer: A Survey of Current Practice across Australia and New Zealand"

_jcm, 2023, doi:10.3390/jcm12062146_

Round 1

Reviewer 1 Report

The current manuscript by Whish-Wilson et al explored clinical management practices before and after surgical lung cancer care in Australia and New Zealand, emphasizing physiotherapy and exercise. The manuscript is very well-written. Authors described the study design, and data analysis elaborately.

Authors should consider addressing the following suggestions.

Suggestions

1.      Authors should consider including one figure comparing pre-operative vs post-operative practices. It is not necessary to include all the observations in this figure. But if authors can identify critical differences, this may be highlighted here.

2.      It is not clear how the specific “Factors influencing respondent management of people with lung cancer” were selected. From table 2 authors concluded “Established workplace practices and personal experience had the most influence on clinicians’ management of people with lung cancer (Table 2.)”.

3.      The title of the study indicates “Physiotherapy” and “exercise management”. For the readers not familiar with this field, it will be very helpful is authors can differentiate these two practices. The abstract mostly focuses on pre- and post-operative exercise regimes. What other physiotherapy practices can be considered here.

4.       Authors are requested to expand the limitations section in the discussion, at their own discretion.

Reviewer 2 Report

This manuscript aims to determine the clinical application of international recommendations on physical exercise prior to- and after- lung resection surgery for lung cancer. The authors conducted a survey all over Australia and New Zealand to determine the diffusion in clinical practice of these indications. Sadly, it is interesting to see that a gap persists between scientific evidence and clinical practice, especially in the case of lung cancer surgery, in which a physical pre-exercise and a fast and complete recovery are essential to reduce the risk of postoperative complications and improve the overall outcome. In particular, patients could undergo a second surgery (doi:10.21037/jtd.2018.07.21) and physiotherapy is essential for a surgery-fit cohort. In fact, physical exercise protocols have been implemented in all types of surgery (doi: 10.3390/app11041513) with evidence (doi: 10.1001/jamasurg.2016.4952).

After the implementation of these concepts, I think that the manuscript is a valuable one.

Reviewer 3 Report

A study of clinical importance in which physiotherapy and pre- and post-operative exercise are important for patients undergoing cardiothoracic surgical resection of lung cancer, but the implementation into routine clinical practice is lacking.

1.     Suggest to include sample size calculation to justify the number of samples (n = 102) to ensure adequate power of the study.

2.     Just wonder why median IQR is used for all data in Table 1 instead of mean. Was the normality assessment done for all continuous data? Is yes, which statistical method was used?

3.     One of the study objectives was to determine the success of implementing the exercise program into routine clinical practice among the studied patients. I am wondering how the success rate was being measured (based on survey alone). What is the cut off take up rate to consider the program being successful?
